# Adults with Trisomy 21 Have Differential Antibody Responses to Influenza A

**DOI:** 10.3390/vaccines10071145

**Published:** 2022-07-19

**Authors:** Stephanie James, Robert C. Haight, Cassandra Hanna, Lindsey Furton

**Affiliations:** 1School of Pharmacy, Regis University, Denver, CO 80221, USA; 2Department of Pharmacotherapy, College of Pharmacy, University of North Texas Health Science Center, Fort Worth, TX 76107, USA; robert.haight@unthsc.edu (R.C.H.); cstroup@regis.edu (C.H.); 3School of Public Health, Boston University, Boston, MA 02118, USA; lefurton@bu.edu

**Keywords:** Down syndrome, vaccine, influenza, Trisomy 21

## Abstract

Down syndrome is caused by an extra copy of chromosome 21. In the past two decades, the life expectancy of individuals with Down syndrome has significantly increased from early 20s to early 60s, creating a population of individuals of which little is known about how well they are protected against infectious disease. The goal of this work is to better understand if adults with Down syndrome are well protected against influenza following vaccination. We obtained plasma samples from 18 adults (average age = 31yo) with Down syndrome and 17 age/gender-matched disomic individuals, all vaccinated against influenza. Antibody concentration to influenza A was measured using ELISA and antibody titers were measured using a hemagglutinin inhibition assay. Statistical analysis was performed using Stata Statistical Software. Adults with Down syndrome had a significantly increased concentration of antibodies to a mixture of influenza A viral proteins; however, they had a significantly decreased titer to the Influenza A/Hong Kong compared to disomic controls. These findings suggest that more vigorous studies of B- and T-cell function in adults with Down syndrome with respect to influenza vaccination are warranted, and that this population may benefit from a high-dose influenza vaccine.

## 1. Introduction

Trisomy 21 (T21; Down syndrome) is a congenital abnormality in which an individual inherits three copies of chromosome 21, with an occurrence rate of 10–14 per 10,000 live births [1,2]. A variety of factors have resulted in the life expectancy of affected individuals increasing from an average 25 years in the early 1980s to over 50 years in the early 2000s [2]; this has yielded an adult population with distinct healthcare needs. These may include, but are not limited to, autoimmune diseases, celiac disease and Alzheimer’s disease. It has also been established that adults with T21 are particularly susceptible to respiratory infections, which remain a primary cause of death in such individuals [3]. A study of 297 hospitalizations of 120 adults with T21 (between the ages of 18–73 years) demonstrated that individuals with T21 have significantly longer and more frequent hospital stays than their disomic (D21) counterparts (referred to herein as disomic adults; D21), with more than a quarter of such hospitalizations resulting from respiratory infections [4]. Data from the 2009 H1N1 influenza epidemic comparing people with T21 to the typical population demonstrated that, despite similar influenza symptom onsets, adults with T21 had a 16-fold increase in hospitalizations, an 8-fold increase in intubation, and a 335-fold increased risk of death [5].

Although there has been a rapid increase in the number of adults living with T21, there is little known about antibody responses to influenza and the influenza vaccine in this population. Studies evaluating in vitro leukocyte proliferation and antibody production in response to influenza antigens have demonstrated a marked decrease in both variables in those with T21 compared to disomic subjects [6,7]. However, such studies have been limited to small sample size with a large age range, making it difficult to draw conclusions about the specific adult response to influenza vaccination. One study did show that adults aged 20–47 have decreased antibody responses to pneumococcal polysaccharide vaccines compared to age- and sex-matched typical adults [8]. Unfortunately, this study did not include an evaluation of influenza antigens. 

Most studies evaluating antibody responses have been in children and have demonstrated a deficiency in antibody production in response to vaccination [6,7,9,10]. In this brief report, we provide a small dataset of adults immunized against influenza and characterize their antibodies to influenza.

## 2. Materials and Methods

Plasma samples from adults with T21 and disomic adults were obtained from the Linda Crnic Center for Down syndrome research at the University of Colorado School of Medicine, along with patient age, gender and immunization history (Table 1). All patients had an influenza vaccination for the 2016–2017 season within 12 months of blood draws.

### 2.1. Antibody Concentration Measurements

Antibody concentration was measured using commercially available kits from Abcam (Cambridge, MA, USA) according to the manufacturer’s instructions. These kits were designed to measure total IgA (cat. Ab108743), IgM (cat. Ab108747) and IgG (cat. Ab108745) to influenza A nucleocapsid and envelope proteins. Additionally, total IgA (cat. Ab108744), IgM (cat. Ab108748) and IgG (cat. Ab108746) against influenza B nucleocapsid and envelope proteins were evaluated. Briefly, plasma samples were diluted 1:200 and incubated for an hour on 96-well microtiter plates coated with influenza nucleocapsid and envelope proteins. Plates were washed, incubated with anti-human antibodies labeled with horseradish peroxidase (HRP) for 30 min, and detected using TMB substrate. All samples were performed in duplicate, and the average Optic Density (OD) was used in statistical analysis. 

### 2.2. Hemagglutination Inhibition Assays (HIA)

The ability of antibodies to inhibit influenza A binding was measured using hemagglutination inhibition assays (HIA), performed by Virapur (San Diego, CA, USA). HIA activity was measured against A\California\07\2009 and A\Hong Kong\4801\2014 X-263B. Briefly, samples were treated with Receptor Destroying Enzyme (RDE) and diluted beginning with a 1:10 dilution and ending with a 1:640 dilution. Diluted samples were tested for antibody ability to interfere with the binding of the hemagglutinin (HA) protein of the designated influenza virus and red blood cells, and titer results reported. 

### 2.3. Antibody Isotyping

Total serum antibody isotypes were measured using the Procarta Multiplex ImmunoAssay for Human Isotyping (Invitrogen, Carlsbad, CA, USA) as per manufacturer’s instructions. Briefly, plasma samples were diluted 1:20,000 and incubated with magnetic beads overnight at 4 °C after which the beads were washed and plasma antibodies were detected using antibodies labeled with Streptavidin-PE. Samples were evaluated using a Luminex instrument at the University of Colorado Cancer Center. All samples were performed in duplicate, and the average concentration was used in statistical analysis. 

### 2.4. Data Analysis and Statistics

This study utilized a cross-sectional design and analyzed results from 35 subjects (51% T21, 49% D21; Table 1). Data from all subjects were entered into an Excel spreadsheet and subsequently imported into Stata Statistical Software (StataCorp, College Station, TX, USA, 2015).

To analyze the effect that T21 has on a patient’s IgG, a multiple variable OLS regression was utilized. The model takes the form of:


Y_i_ = β_0_ + β_1_T21_i_ + β_3_Female_i_ + ε_i_
(1)


This model represents Y as the antibody level for individual i; T21 is an indicator variable if the individual has T21; Female is an indicator variable if the individual is female; and ε is the stochastic term.

This analysis allows for the control of age and gender while comparing the difference of Y between T21 and D21. Models (1)–(3) analyzed IgM, IgA and IgG antibody concentrations specific to influenza A, respectively utilizing an OLS regression (Table 2). Models (4) and (5) examined antibody titers specific to influenza A strains Hong Kong and California, respectively utilizing a quartile regression (Table 3).

## 3. Results

### 3.1. Antibody Concentrations 

The concentration of antibodies to both influenza A and B were measured using a colorimetric ELISA. An increased concentration of antibodies was reflected by an increased OD at 450 nm. Samples from individuals with T21 had significantly increased concentrations of IgG antibodies against influenza A (*p* < 0.05, Table 2, Figure 1), but no statistically significant differences in total IgM or IgA against influenza A. No differences of antibody concentrations against influenza B (data not shown) were observed.

### 3.2. Hemagglutination Inhibition Assays

To evaluate if the antibodies to influenza A were able to inhibit viral activity, we used a hemagglutination inhibition assay. This assay was performed by Virapur, Inc., using strains A\Hong Kong and A\California, both of which circulated during the 2016–2017 influenza season and were included in that year’s vaccine preparation. Overall, adults with T21 had a significantly lower antibody titer to A\Hong Kong compared to disomic adults (*p* = 0.05). This assay did not differentiate antibody isotypes. Further analysis of dividing the samples into quartiles demonstrated that the significant differences were due to differences at the highest titers (Table 3). That is to say, the samples from individuals with T21 never developed high titers compared to typical individuals. No significant differences were observed between groups to A\California (Figure 2).

### 3.3. Antibody Isotyping 

The total concentrations of antibody isotypes IgA, IgM, IgE and IgG1-4 were measured from plasma samples. No significant differences between groups for each isotype were observed (data not shown). 

## 4. Discussion

Respiratory infection is a leading cause of death in adults with T21 [11,12,13,14]. The infectious agents responsible for the causes of respiratory disease may vary and are not fully elucidated in the Trisomy 21 literature. Influenza is a common respiratory infection resulting in thousands of deaths each year [15]. This virus may also predispose individuals to secondary bacterial infections causing pneumonia [16,17,18]. The strains of influenza vary year to year; during the time frame of this study, the majority of circulating virus was influenza A, primarily of the H3N2 type and similar to the A/Hong Kong type. The formulation of influenza vaccines for the 2016–2017 season included A/California/7/2009 (H1N1) pdm09-like virus and A/Hong Kong/4801/2014/(H3N2)-like virus [19]. Thus, individuals who received the 2016–2017 vaccine should have developed a protective antibody response to seasonal influenza. 

Data presented in this report suggest a differential antibody response to the influenza vaccine in adults with Trisomy 21 compared to disomic adults. Interestingly, adults with T21 had a greater concentration of IgG antibodies to influenza A compared to disomic adults. However, the assay employed tested antibodies against multiple influenza A nucleocapsid and membrane proteins and could not differentiate between antibodies raised to immunization and those resulting from natural influenza infection. Unfortunately, the date of immunization was not collected by the biobank from where samples were obtained. Hence, the greater OD may have been due to antibodies generated in response to vaccination (as all subjects had received the 2016–2017 seasonal influenza vaccine within 12 months), or they may have been generated from an actual influenza infection. Unfortunately, patients were not questioned about any recent respiratory infections. 

To better evaluate if adults with T21 were able to develop a specific antibody response to block the influenza virus, we utilized a hemagglutinin inhibition assay. This assay is the gold standard to evaluate the antibody titer to influenza. Interestingly, although adults with T21 had greater concentrations of IgG antibodies against influenza A, they had a significantly decreased titer to influenza A (A/Hong Kong) compared to age- and gender-matched disomic adults. This suggests that, while adults with T21 can create antibodies to influenza A, they may not be as of high avidity compared to disomic adults. It may be hypothesized that, despite vaccination, adults with T21 are unable to develop a reliable response to vaccine antigens, which leaves them susceptible to clinical or subclinical infections compared to D21 counterparts.

The lack of high avidity antibodies in individuals with T21 may be due to abnormalities in B- (or T)-cell compartments. Several past studies have shown abnormalities in the lymphocyte populations from individuals with T21. B-cell populations, including memory B-cell types, are reduced in children with T21 [20,21]. A notable study by Valentini et al. (2015) demonstrated that, in children with T21, influenza vaccination did not result in as many memory-switched B cells compared to their disomic counterparts [9]. T-cell populations are also disproportionate in children with T21 compared to their disomic peers [22,23,24]. A defect in CD4 T-cell proliferation in response to antigen has been reported in children with T21, which may contribute to a defect in T-helper responses to vaccination [25,26]. Thus, deficits in antibody efficacy may be due to abnormal T–B-cell interactions or B-cell abnormalities, including abnormal somatic hyper mutation and affinity maturation. B-cell responses are also driven by cytokine action. Indeed, an abnormal cytokine profile has been shown in those with Down syndrome [27,28]. The IFN-alpha receptor is located on chromosome 21 and has been shown to be increased in expression and functionality in those with Down syndrome [26]. This may be particularly relevant to antibody production as IFN-alpha is known to directly modulate antibody production [29], with excess IFN-alpha potentially decreasing antibody synthesis.

Finally, it may be suggested that, because of a less effective antibody response, adults with T21 do not efficiently eliminate infectious pathogens, such as influenza. This may result in the continuous production of antibodies (of lower quality) and thus explain why these individuals have a higher level of antibodies to the influenza virus, as shown in our study, although they are not of sufficient avidity to fully avoid pathology and disease.

## 5. Conclusions

Because respiratory infections are a leading cause of morbidity and mortality in adults with T21, vaccination against such infections are important to the health and well-being of these individuals. More research needs to be performed to further elucidate the role of B- and T-cell interactions, B-cell switching and affinity maturation in adults with T21 to ensure they are protected from vaccine-preventable diseases. In this paper, we suggest that adults with T21 may not be fully protected from influenza A despite vaccination and may need to be considered for a high-dose influenza vaccine. Furthermore, it may be advised to check antibody titers to other vaccine-preventable diseases to ensure those individuals with Down syndrome are protected in the event of an outbreak or exposure. 

## Figures and Tables

**Figure 1 vaccines-10-01145-f001:**
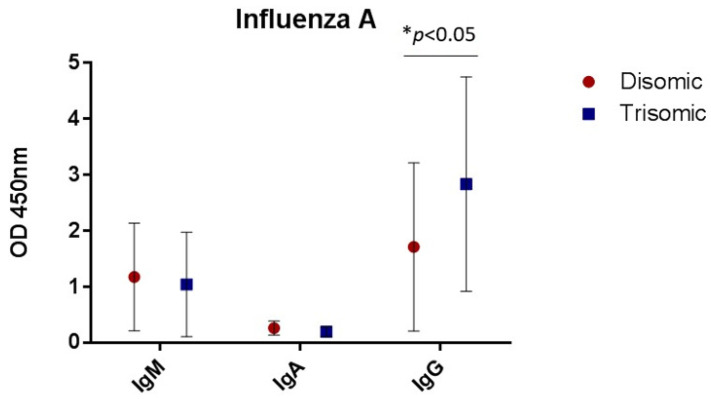
Using an ELISA to evaluate antibody concentration against influenza A virus. Results indicate that individuals with T21 have a significantly higher concentration of IgG antibodies to influenza A compared to typical, disomic individuals.

**Figure 2 vaccines-10-01145-f002:**
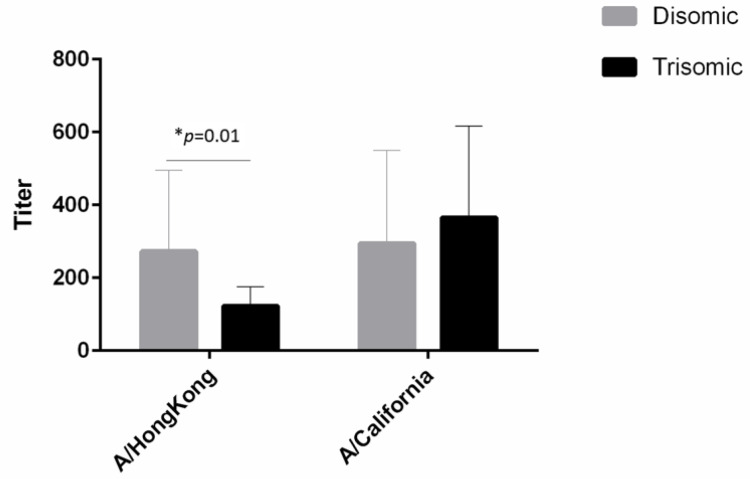
Antibody titers were evaluated using the hemagglutinin inhibition assay. Antibodies to Influenza A/Hong Kong from individuals with Trisomy 21 were significantly decreased compared to disomic individuals.

**Table 1 vaccines-10-01145-t001:** Population Demographics.

	Disomic Individuals	Trisomic Individuals
**N**	17	18
**% Female**	76.47	61.11
**% Male**	23.53	38.89
**Age [mean(sd)]**	37.32 (13.23)	30.91 (10.36)

**Table 2 vaccines-10-01145-t002:** MLS Regression on the effects of T21 on Antibody Concentrations.

	IgM (1)	IgA (2)	IgG (3)
Trisomy 21	−0.20 (0.31)	−017 (0.14)	1.21 * (0.58)
Female	−0.29 (0.34)	−0.14 (0.15)	−0.43 (0.63)
Constant	1.42 *** (0.34)	0.55 *** (0.15)	1.89 ** (0.64)
Observations	36	36	37
Adjusted R-squared	−0.03	0.00	0.09

The table above shows the results from the multiple variable regression. This analysis controls for gender (standard errors in parenthesis; * *p* < 0.05 ** *p* < 0.01 *** *p* < 0.001).

**Table 3 vaccines-10-01145-t003:** Quartile Regression on the effects of T21 on Antibody Concentrations.

	A/Hong Kong (4)	A/California (5)
20th Quartile		
Trisomy 21	−40.00	120.00
Female	−80.00	20.00
Constant	160.00 **	20.00
40th Quartile		
Trisomy 21	0.00	80.00
Female	0.00	−80.00
Constant	160.00	160.00
60th Quartile		
Trisomy 21	0.00	320.00
Female	0.00	160.00
Constant	160.00	160.00
80th Quartile		
Trisomy 21	−480.00 *	0.00
Female	−0.00	0.00
Constant	640.00 ***	640.00 ***

The results of a quartile regression indicating that among all individuals with A/Hong Kong antibody titer in the 80th percentile; those with Trisomy 21 had significantly lower titers than those without Trisomy 21 (* *p* < 0.05 ** *p* < 0.01 *** *p* < 0.001).

## Data Availability

Not applicable.

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
