# Peer review of "Adults with Trisomy 21 Have Differential Antibody Responses to Influenza A"

_vaccines, 2022, doi:10.3390/vaccines10071145_

Round 1

Reviewer 1 Report

The manuscript entitled "Adult with Trisomy21 have differential antibody responses to Influenza A' is good to read and have some novel observations that will enrich the science of immune response among individuals with Down syndrome. This reviewer has some suggestions for minor corrections and clarification in the manuscript.

1. The major flaw in the view of this reviewer is absence of clear mention of time gap or incubation period between vaccination and measurement of antibody titre. The time is an important variable factor for determination of immune response following vaccination. So it is desirable that authors should provide a detail regarding this information may be in separate table. 

2. In the figure 2 it is not clear whether authors have measured the titre of IgG or collective titre of all antibody isotypes. 

3. In discussion author stated that they observed elevated elevated level of antibody response against influenza vaccine in T21 than D21. But the reason behind that is not properly anticipated. This observation raised question in the mind of this reviewer. In covid-19 vaccination study this reviewer observed low antibody titre among T21 than D21. So, present observation is interesting. In this regard the time gap between vaccination and measurement is pivotal. It is possible that the average time gap for T21 is shorter following vaccination than for D21. If so, authors may observed the present result. Otherwise, it may be the vaccination specific response which is less likely. So, it is recommended to give a clear picture regarding interval between vaccination and measurement. 

4. Weak avidity in T21 may be the result of aberrant cytokine profile which is base cause of immune immune dysregulation in T21. It will be better to highlight the role of cytokines in avidity(with proper citation) and then suggest probable effect in T21

Author Response

Dear Reviewer,

Thank you for your thoughtful comments and suggestions.  In response:

  1. We agree that the timetable of vaccination is very important and would love to include this information.  However, this information (date of the vaccine) was not collected and provided to us by the biobank.  While very unfortunate, we still believe that the data is worth reporting for other investigators.
  2. Thank you, we added a sentence in the results section that the assay did not differentiate between antibody isotypes. 
  3. Thank you, a very interesting finding.  Our study did find a lower titer as well, it was just the total antibody that was higher.  One suggestion is that due to lower titer, individuals with T21 may have low level infection which could account for a higher antibody concentration.  
  4. Thank you, I think the question in #3 ties in to the point about cytokine responses and in particular IFN alpha.  IFN-alpha plays a direct role on Ig synthesis according to Peters and is now mentioned/discussed in the conclusion.

We thank you very much for reviewing our paper.

Reviewer 2 Report

This MS addresses the issue of response to vaccination in adults with Down syndrome.
The authors focused on Influenza vaccination. In order to see if any difference is achieved comparing patients with T21 with normal disomic individuals, the authors used two age and sex matched groups, one with T21, and the second one as control disomic individuals. No difference resulted regarding influenza B vaccination. Instead, the antibody titer against influenza A Hong Kong strani was statistically significantly decreased in adults with T21. Tha coerente meaning of this result is the insufficient protection against influenza A virus, Hong Kong strain. The authors suggested to use high vaccine dose in those adults with T21. 

This is a well-designed brief study highlighting the low response of individuals with T21 to influenza A vaccination. I do not have specific suggestions of improvement.

Author Response

Thank you very much for reviewing our paper.

Reviewer 3 Report

The manuscript “Adults with Trisomy 21 have differential antibody responses to Influenza A” is a good review with detailed information about how adults with T21 may not be fully protected from influenza A despite vaccination and may need to be considered for the high dose influenza vaccine. It also alerts and suggests to check the antibody titers to other diseases that can be prevented by vaccination to ensure those individuals with Down syndrome are protected in the event of an outbreak or exposure. This manuscript may be helpful for the scientific community. Although the manuscript contains data for publication, but there are few suggestions to improve this work as given below.

Specific Comments:

  1. The abstract should be revised to more clearly define the manuscript's goal and eliminate any gaps.
  2. To highlight the significance of studying this genetic condition, it would have been preferable if the introduction had facts on the risk factors regarding health problems associated with T21.
  3. The authors need to clearly demonstrate the relevance and impact of their review.
  4. The reviewer omitted to display some data that represented the unfavourable findings. For instance, line 112 and 126 (data not shown). It might have been included as well to give the review paper additional depth.
  5. Please revise the article to include clearer phrases, rectify the errors in the citation, and eliminate unnecessary ambiguity or incompleteness.

Minor comments

1. Please rephrase the statements such that they are easier to grasp. For instance, Lines 133 and 177 are similar and should be rephrased.

2.      Heading for Table – 1 is missing. To make it easier to understand, correctly label it and include a legend for the same.

3.      There are no legends for the figures. Please include accurate legends for figures 1 and 2.

4.      Line No. 27 – Insert space after T21.

5.      Line No. 161 – Remove the bracket.

6.      Place a full stop after the citation, not before. Take this into account for the manuscript's remaining portions as well.

Although the manuscript's English is fairly well, certain edits, such as those for the misspelt words, grammatical errors, and restructuring, are necessary. For example, the spelling of “Cambridge” needs to be corrected in line 57 and line 65 needs grammar correction - the average OD was used in statistical analysis. Please check the manuscript for any other such errors.

Author Response

Hello,

Thank you very much for reviewing our paper and your suggestions.  In response:

  1. We added a sentence in the abstract stating the overall goal of the work.
  2. Thank you, we also highlight some other diseases to which adults with DS are particularly susceptible.
  3. Thank you, to increase relevance and impact we discuss the potential role of IFN alpha in the Discussion.  Several papers have shown the increase in IFN-alpha expression and functionality which may play an important role in antibody production.
  4. We could add in a figure regarding Influenza B if reviewers deem it necessary.  We opted not to include this because the findings were not significant, and also because the predominant strain of influenza was of the A lineage.
  5. To rectify any grammatical errors the authors carefully reviewed the manuscript and have also requested an editor for our writing services to review.
  6. Table headings and all figure legends have been included.  Our sincere apologies this was left out.

Thank you again for the review and excellent  suggestions.